# Insights from the 2018 IPC Benchmarks

**Isabel Cenamor** and **Alberto Pozanco**

Departamento de Informática, Universidad Carlos III de Madrid
Avda. de la Universidad, 30. 28911 Leganés (Madrid). Spain
icenamorg@gmail.com, apozanco@pa.inf.uc3m.es

## Abstract

The International Planning Competition (IPC) empirically evaluates state-of-the-art planning systems on a set of benchmark problems. The selection of this benchmarks plays an important role in the competition, since they can significantly affect competition results.

In this paper we analyze the diversity of the benchmarks employed in the last IPC through extracting some features from the domains and problems of the optimal track. Finally, we provide some insights from the collected data and propose to use a similar method to select the benchmarks of future competitions.

## Introduction

In Artificial Intelligence, it is common to have competitions associated with each particular research area. These competitions aim to bring together different state-of-the-art systems, evaluating them on a set of benchmarks. Just like in Satisfiability Testing (Järvisalo et al. 2012), or in Answer Set Programming (Gebser, Maratea, and Ricca 2017), the Automated Planning community promotes the development of innovative planning techniques since 1998 through the International Planning Competition (IPC).

In the IPC, participating planning systems are tested in several benchmark problems. The selection of these benchmark domains and problems instances plays an important role in the competition, since they can significantly affect competition results (Howe and Dahlman 2002). This task is non-trivial, and it has given a lot of headaches to the organizers of previous competitions (Linares López, Celorio, and Olaya 2015; Vallati, Chrpa, and McCluskey 2018). One of the main consensuses among the different post-competition discussions, is that benchmark domains and problems should be as diverse as possible (Vallati and Vaquero 2015), in order to (1) enrich the competition, and (2) not bias the results in favour of any planner.

In this paper, we analyze the diversity of benchmarks employed in the IPC 2018. We do that by extracting some features from the domains and problem instances of the optimal tracks. Features from domains and problems have been successfully used to predict planner's coverage (Roberts et al.

2008; Roberts and Howe 2009) or run time (Fawcett et al. 2014); and also to generate state-of-the-art planning portfolios (Cenamor, de la Rosa, and Fernández 2016). Here we use these features to evaluate and analyze the diversity of the competition benchmarks.

In the rest of the paper we introduce the feature extraction process, including a brief description of the features. Then, we detail how we process the raw data and introduce our different analyses, in which we also include the IPC 2014 for comparison purposes. Firstly, we perform an intra-domain analysis to test how diverse are the problem instances within the same planning domain. Secondly, we perform an inter-domain analysis to test how diverse are the domains and problems among them. Finally, we merge the data from IPC 2014 and IPC 2018 to group the domains and problems based on their similarity. We conclude our analysis by providing some insights from the results, and outlining a procedure similar to the one we carried out to select the benchmarks of future IPCs.

## Planning Features

We use the same features extracted by the IBaCoP family of portfolios (de la Rosa, Cenamor, and Fernández 2017). The extraction process collects data from different steps of the Fast Downward system (Helmert 2006), in the version that was available before the IPC 2014. We briefly describe the set of 114 real-valued features we will use throughout our analysis by classifying them into the following categories:

- **PDDL**. These features are extracted from the original domain and problem definition in the PDDL files. If the domain contains conditional effects, we parse them using ADL2STRIPS (Hoffmann et al. 2006). Specifically, we have implemented the compilation that creates artificial actions for effect's evaluation (Nebel 2000). Some of these features are: number of actions, number of objects or number of goals.

- **Fast Downward Instantiation**. The pre-processor of Fast Downward instantiates and translates the planning tasks into a finite domain representation (Helmert 2009). Some of these features are: number of mutex groups, memory used for the translation process or whether action costs are used or not.

- **SAS$^+$**. These features are based on the causal graph (Helmert 2004) and domain transition graphs (Jonsson and Bäckström 1998) associated to the finite domain representation. Some of these features are: number of variables and edges of the causal graph, ratio of variables involved in the goal, or sum of the number of nodes of all domain transition graphs.

- **Heuristics**. These features represent different heuristic values of the initial state of the search. Some of these features are: the FF heuristic (Hoffmann and Nebel 2001), the landmark-cut heuristic (Helmert and Domshlak 2009) or the red-black heuristic (Katz, Hoffmann, and Domshlak 2013).

- **Fact Balance**. These features are extracted from the relaxed plan of the initial state when the FF heuristic is computed.

- **Landmarks**. These features are extracted from the landmark graph computed by Fast Downward (see details in (Cenamor, de la Rosa, and Fernández 2016)). Some of these features are the number of landmarks, the number of edges in the landmark graph or the number of intermediate nodes in the graph.

Through extracting these features, we aim to characterize each problem instance to later compare them.

## Data Extraction

We extract the IBaCoP features of the domains and problems of the IPC 2018 optimal track[1]. We decided to perform our analysis on that track given that we were able to successfully extract most of the features for all the problem instances, while the extraction results were worse in the satisficing track due to time and memory issues. We also extracted the features of the IPC 2014 optimal track[2] so we can compare them properly. The feature extraction process was run on an Intel Core i5-2410M CPU @ 2.30GHz and 4GB of RAM. We apply a time limit of 1800 seconds to the extraction of the features of each problem[3].

|  | #Features | Success |
|---|---|---|
| PDDL | 8 | 100% |
| FD | 16 | 100% |
| SAS$^+$ | 50 | 100% |
| Heuristic | 16 | 82% |
| FB | 10 | 76% |
| Landmarks | 14 | 100% |
| Total | 114 | |

Table 1: Feature type, number of features per type, and extraction success.

[1]https://bitbucket.org/ipc2018-classical/domains/src

[2]https://helios.hud.ac.uk/scommv/IPC-14/benchmark.html

[3]The extracted data is available at https://github.com/apozanco/wipc-icaps2019 for IPC 2014 and 2018

| Name | Min | Max | Mean | Std | Median |
|---|---|---|---|---|---|
| agricola | 32.0 | 164.0 | 87.9 | 36.2 | 80.0 |
| caldera | 97.0 | 382.0 | 319.8 | 76.6 | 339.0 |
| caldera-split | 64.0 | 280.0 | 135.3 | 50.0 | 124.0 |
| data-network | 2.0 | 6.0 | 3.6 | 0.9 | 3.0 |
| nurikabe | 61.0 | 634.0 | 311.1 | 169.1 | 324.5 |
| organic-synthesis-split | 80.0 | **844.0** | 257.8 | **230.8** | 133.0 |
| organic-synthesis | - | - | - | - | - |
| settlers | 100.0 | 283.0 | 177.6 | 52.1 | 167.5 |
| snake | 12.0 | 53.0 | 29.5 | 13.1 | 26.0 |
| spider | **167.0** | 671.0 | **383.6** | 137.3 | **408.0** |
| termes | 2.0 | 3.0 | 2.2 | 0.4 | 2.0 |
| petri-net | 7.0 | 30.0 | 18.4 | 6.8 | 17.5 |

Table 2: Minimum, maximum, average, standard deviation and median time to extract features for each domain in the IPC 2018 optimal track. In bold the higher values per column.

Table 1 shows the extraction success for each feature type in the IPC 2018. As we can see, most of the features are extracted correctly. Table 2 shows different metrics related to the time needed to extract the features in each domain of the IPC 2018. While it is easy to extract the features in some domains such as termes and data-network, there are other domains like spider or organic-synthesis-split in which this process may take up to two more orders of magnitude. This is because these domains present ADL, action costs, and negative preconditions, which need a special PDDL pre-process. We discarded organic-synthesis, since we only could extract the features of 5 problem instances within the time limit.

## Data Pre-processing

After extracting the features, we have a features matrix $\mathcal{M}$. Each row in the matrix represents a problem instance $p_k$, and each column represents a feature $f_i$. Each cell contains the numeric value of a feature for that problem, $f_i(p_k)$. As we showed in Table 1, we do not have all the features' values for all the problems. So first of all, we have to deal with the missing values.

Here we have two main options: (1) discard those features with any missing value for any problem; or (2) substitute the missing values by actual values. Discarding features implies losing information. If the system is not able to extract a feature in a problem, it means that this instance is different from others in which it can be extracted. Moreover, we have high extraction success in almost all the features, so we opted for the second alternative, substituting the missing values. These values can be replaced in many ways. We chose to replace them by either:

- Setting the feature value to 0, if there is no problem in the domain in which the feature has been successfully extracted.

- Setting the feature value to the average of that feature values in the domain, if there exist at least one problem in the domain in which the feature has been successfully extracted.

Then, we cleaned the data by removing the features that were not sufficiently informative. For this purpose, we deleted the set of features which have the same value for all the problem instances of the competition. This make our set of features to reduce from 114 to 107. We deleted 3 features from the PDDL description, 3 from the $SAS^+$ representation, and 1 from the heuristic values.

Finally, we normalized the features matrix by applying the following equation to every remaining feature $f_i \in \mathcal{M}$,

$$f'_i(p_k) = \frac{f_i(p_k) - f_{min}}{f_{max} - f_{min}}$$

where $f_i(p_k)$ is the current value of the feature $f_i$ in the problem instance $p_k$; $f_{min}$ and $f_{max}$ are the minimum and maximum values of the feature $f_i$ for all the problem instances $p_k \in \mathcal{M}$; and $f'_i(p_k)$ is the new normalized value of the feature $f_i$ in the instance $p_k$. After this process, the features matrix $\mathcal{M}$ is normalized, with all the features' values within the $[0, 1]$ range.

If we take a look to these features' values, there are some that have similar values for all problem instances, while others are very different. As instance, in the IPC 2018, all the problem instances have a similar ratio between the total number of variables and the total number of edges in the causal graph. On the other hand, the most different features correspond with the number of predicates and types in the problem instances.

In the following experiments, we will refer as problem features' vector $\mathcal{V}_{p_k}$ to the list of values that describe the features of a problem instance $p_k$.

## Intra-domain Analysis

Our first analysis aim to test how diverse are the problem instances within the same planning domain. For each planning domain, we compute a matrix with the columns and rows being the problem instances $p_k$ of that domain. Each cell of the matrix denotes the difference between two problem features' vectors $\mathcal{V}_x$ and $\mathcal{V}_y$. This difference is computed as follows:

$$\mathcal{V}_x - \mathcal{V}_y = \sum_{i=1}^{i=107} |f_i(p_x) - f_i(p_y)|$$

This value can range from 0 to 107. Values closer to 0 mean that the two problem instances are similar, while higher values mean diverse problem instances.

To test how diverse the problems of a domain are, we then sum all the rows (or columns) in the matrix and divide that number by $n^2$, where $n$ is the number of problems in the domain. This number reflects how different/similar is an average problem with the rest of problem instances of its domain. This number can range from 0 to 37.5 in the case of domains with 20 problems. Table 3 show the results of our intra-domain analysis for both IPC 2014 and 2018.

As we can see, there is a lot of variation in the results. Domains like parking and visitall in 2014, and spider or organic-synthesis in 2018 have very diverse problems, while all the instances in data-network or barman seem to be similar. The IPC 2018 has more diverse problems within the

| Domain 2014 | Difference | Domain 2018 | Difference |
|---|---|---|---|
| parking | 9.5 | spider | 12.5 |
| tetris | 8.5 | organic-synth | 10.5 |
| visitall | 7.4 | nurikabe | 9.9 |
| transport | 6.7 | caldera | 7.5 |
| tidybot | 4.9 | caldera-split | 4.4 |
| openstacks | 4.8 | petri-net | 4.1 |
| citycar | 4.1 | agricola | 4.1 |
| cave-diving | 4.1 | snake | 3.8 |
| hiking | 3.6 | settlers | 2.2 |
| GED | 3.5 | termes | 1.5 |
| child-snack | 2.2 | data-network | 0.9 |
| floortile | 1.4 | | |
| maintenance | 1.1 | | |
| barman | 0.9 | | |

Table 3: Intra-domain Analysis.

same domain, with an average difference of 5.6 against the average difference of 4.5 in the case of the IPC 2014.

To better illustrate how diverse the problems within a domain are, we plotted together all the problem features' vector of each domain. The results for the domains with most and least similar problems in the IPC 2018 are shown in Figure 1.

We also ran a small experiment to see if these results correlate with the planners' performance. We hypothesized that in domains with similar problems such as data-network, planners would perform similarly, i.e., they would solve most or almost none of the problems in the domain. On the other hand, in domains with different instances such as spider, planners would solve the problems in a more different way. To check this, we computed the standard deviation of each planner solving the problems of each domain (1 if a problem is solved, 0 otherwise). Lower values for a planner imply that it has been able to solve most or almost none of the instances in the domain. Then we compute the average of each planner for each domain.

However, our hypothesis is not met. As instance, spider which is the domain with most different instances, has a standard deviation of 0.47, while data-network which is the domain with most similar problems, has a standard deviation of 0.49. Some of the possible reasons why these results do not correlate are: (1) the competitor planners are very different from each other, and hence some domains and problems could be more suitable for one or other planner; and (2), the fact that a domain has similar problems does not necessarily imply that they can be solved in the same way. This little differences may come from increasing the number of objects, and therefore planners will only solve a small subset of them.

## Inter-domain Analysis

Our second analysis aim to test how diverse are the domains among them. For each planning domain, we compute a domain features' vector $\mathcal{V}_{D_k}$ which is a problem features' vector representative of the domain $D_k$. We do that by assigning to each feature the average of the values of that feature in all the problem instances of the domain.

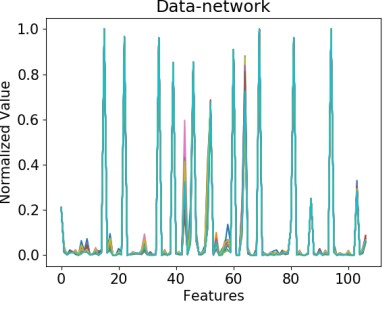

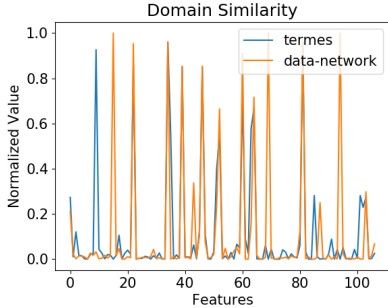

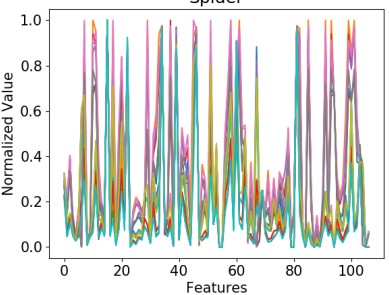

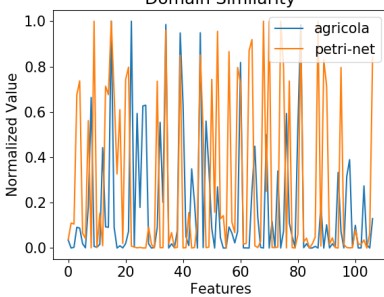

Figure 1: Domains with most similar (top) and least similar (bottom) problems. Each color represent one of the 20 different problem features' vector of each domain.

Figure 2: Similar domains (top) and different domains (bottom) in the IPC 2018.

$$\forall f_i \in \mathcal{M}, f_i(\mathcal{V}_{D_k}) = \frac{\sum_{p_k \in D_k} f_i(p_k)}{|D_k|}$$

By doing this, we are capturing all the information of a domain within just one features' vector. However, we may lose some information, mostly in those domains with diverse problem instances.

After that, we compute a matrix that in this case will have domain features' vectors both in the rows and columns. Each cell of this matrix denotes the difference between two domain features' vectors. This value can range from 0 to 107. If we sum each row (column) in the matrix and divide that number by $k$, the number of domains in the competition, we get how different is on average a domain with respect to the other domains. Table 4 shows the results of our inter-domain analysis of the IPC 2018.

As we can see, petri-net-alignment is the domain that keeps more differences with respect to the rest of domains, with an average value of 27.2. Moreover along with agricola, they are the most different pair of domains. On the other hand, caldera is the domain which is more similar to the others in the competition, with an average value of 12.0. Caldera and caldera-split is the most similar pair of domains in the IPC. This make sense, since both domains only differ in the problems' grounding. To better illustrate how diverse are the domains among them, we plotted together some domain features' vectors together in Figure 2.

We performed the same inter-domain analysis for the IPC

2014. In this case, the most different domain is tidybot, with an average value of 21.4. On the other hand, barman is the domain which is more similar to the others in the competition, with an average value of 9.2.

These maximum and minimum values are less distanced than the values of the IPC 2018. In fact, while the average of the domains' differences in the IPC 2018 is 15.6, this average is 12.0 in the case of the IPC 2014. This means that the set of domains and problem instances in the IPC 2018 is more diverse that the one of the IPC 2014.

We also ran a small experiment to see if these results correlate with the planners' performance. We hypothesized that planners would perform akin in similar domains and different in domains with different features. To check this, we computed for every planner the difference in absolute value of the number of problems solved in each pair of domains. Then we sum the results of each planner for each combination of domains and divide it by the number of planners. Lower values imply that the planners of the competition solve a similar number of problems in the given domains.

In this case, our hypothesis is met in most cases. As instance, if we take termes and data-network (the most similar domains except for the two versions of caldera), we get a value of 0.16, while in the case of agricola and petri-net (the most different domains), we get a value of 0.29. This is a common trend across domains, although there are some cases in which it is not fulfilled. As instance, the value obtained when comparing nurikabe and organic-synthesis is 0.13, which is lower than in the case of termes and data-network. Again, this can happen for the same reasons de-

| | organic-synths | agricola | caldera-split | spider | termes | data-network | snake | nurikabe | caldera | petri-net | settlers |
|---|---|---|---|---|---|---|---|---|---|---|---|
| organic-synth | 0.0 | 19.4 | 14.2 | 20.4 | 17.6 | 15.6 | 16.9 | 15.3 | 13.0 | 33.3 | 11.9 |
| agricola | 19.4 | 0.0 | 16.7 | 22.9 | 18.1 | 13.8 | 17.2 | 15.5 | 14.4 | 33.5 | 15.3 |
| caldera-split | 14.2 | 16.7 | 0.0 | 16.8 | 10.3 | 10.7 | 13.4 | 12.9 | 7.7 | 26.9 | 8.3 |
| spider | 20.4 | 22.9 | 16.8 | 0.0 | 20.7 | 22.1 | 16.2 | 19.3 | 15.7 | 33.2 | 18.3 |
| termes | 17.6 | 18.1 | 10.3 | 20.7 | 0.0 | 8.3 | 11.1 | 14.0 | 11.5 | 25.0 | 12.4 |
| data-network | 15.6 | 13.8 | 10.7 | 22.1 | 8.3 | 0.0 | 13.9 | 13.1 | 10.6 | 25.8 | 10.2 |
| snake | 16.9 | 17.2 | 13.4 | 16.2 | 11.1 | 13.9 | 0.0 | 13.1 | 10.4 | 31.7 | 13.7 |
| nurikabe | 15.3 | 15.5 | 12.9 | 19.3 | 14.0 | 13.1 | 13.1 | 0.0 | 10.7 | 31.3 | 10.8 |
| caldera | 13.0 | 14.4 | 7.7 | 15.7 | 11.5 | 10.6 | 10.4 | 10.7 | 0.0 | 29.2 | 8.9 |
| petri-net | 33.3 | 33.5 | 26.9 | 33.2 | 25.0 | 25.8 | 31.7 | 31.3 | 29.2 | 0.0 | 28.9 |
| settlers | 11.9 | 15.3 | 8.3 | 18.3 | 12.4 | 10.2 | 13.7 | 10.8 | 8.9 | 28.9 | 0.0 |
| Average | 16.1 | 17.0 | 12.5 | 18.7 | 13.5 | 13.1 | 14.3 | 14.2 | **12.0** | **27.2** | 12.6 |

Table 4: Differences among domains from the IPC 2018. Green cells identify diverse domains, while purple cells identify similar domains. Bold numbers represent the most diverse and similar domains in the competition.

scribed in the intra-domain analysis.

## Clustering Domains

Our last analysis aim to group the benchmarks of the IPCs 2014 and 2018 based on their similarity. For this purpose, we merge the raw data of the extracted feaures of both competitions, and follow the same pre-processing step as before. We compute a features' vector for each domain, as we did in our inter-domain analysis.

Now we perform a hierarchical clustering to the 25 domain features' vectors (11 from the IPC 2018 and 14 from the IPC 2014). We do that to test (1) if there exist similar domains across different competitions, hence being part of the same cluster; and (2) which domains are the most different from the rest, hence conforming they own cluster.

Figure 3 shows the result of our hierarchical clustering in the shape of a dendrogram. As we can see, domains like barman, child-snack or hiking are grouped together first. This means that they are the most similar ones within both competitions. The most diverse domains are shown at the bottom of the y axis. They correspond to spider, settlers, visitall, agricola, parking, tidybot, organic-synthesis and petri-net-alignment, which is the most different domain across competitions.

## Discussion

The selection of the benchmark domains and problem instances plays an important role in the IPC. A desirable property of these benchmarks is that they should be as diverse as possible, in order to enrich the competition and not bias the results in favor of any planner. In this paper we have presented a study of the diversity of the benchmarks of the IPCs 2018 and 2014. We carried out three different analyses: an intra-domain analysis, to test how diverse are the problem instances within the same planning domain; an inter-domain analysis, to test how diverse are the domains and problems among them; and a clustering procedure to group the domains of both IPCs based on their similarity.

Our analyses suggest that the IPC 2018 employed more diverse domains and problem instances than the IPC 2014. From the results, we can also conclude that in both competitions there are domains which are not similar to any other,

not only within the same competition but also if we take other IPCs into account. We think these *different* domains such as spider, agricola or parking really enrich the IPC.

However, our results should be read carefully, and more like a photograph of the benchmarks, than a test that determines how good or bad a problem/domain/competition is.

The first reason for that is that throughout our analyses, we measure the similarity or diversity of problems and domains with respect to their set of features. These features, even though proved useful by other works, may not conform the best set of features for differentiating problems; also, some of these features may be too correlated and introduce noise in the similarity computation. Further work on the set of features should be done to properly characterize problem instances. We also want to note that the fact that a domain has similar problem instances, or a competition similar domains, does not mean anything bad. It may be the case that all these similar domains are challenging for the planners. Moreover, the low intra-domain differences in domains like barman may be related to having problems with increasing number of objects or goals. These type of domains are useful to test planners' scalability and should be present at future competitions.

The second reason is that a competition comprises both benchmarks and planners. Although other works has focused on that relationship (Cenamor, de la Rosa, and Fernández 2016; de la Rosa, Cenamor, and Fernández 2017), here we only focused on the benchmarks, leaving the planners' performance over these benchmarks out of the scope of this paper. This work should be extended to take diverse planners into account, characterizing them and analyzing how they solve each kind of domain and/or problem instances. By doing this, it would be possible to know which set of features make the problem instances hard to solve by each kind of planner. This information would be very useful when selecting the domains and problems of a competition.

We believe that by improving this work in the outlined directions, we may have some of the key ingredients to select (or even generate) diverse benchmarks for future IPCs.

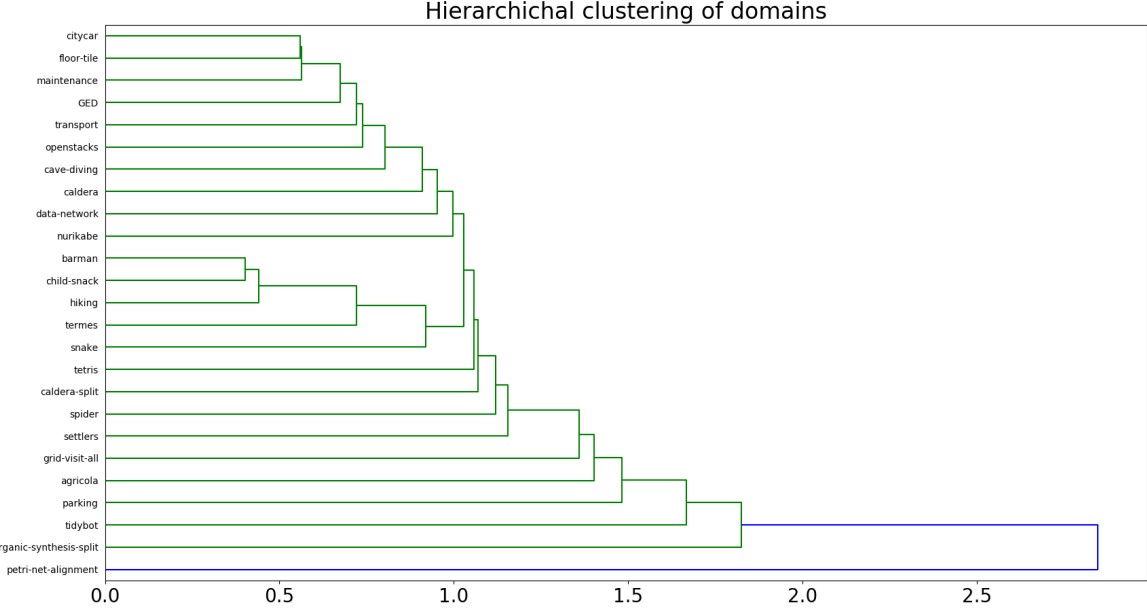

Figure 3: Hierarchical clustering of domains. The domains are represented in the y axis, while the x axis represents a measure of error. Domains grouped first are the most similar. Domains grouped last, depicted at the bottom of the y axis, are the most different.

## Acknowledgments

Alberto Pozanco is funded by FEDER/Ministerio de Ciencia, Innovación y Universidades Agencia Estatal de Investigación/TIN2017-88476-C2-2-R and RTC-2016-5407-4.

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
