# OpenReview forum: "Insights from the 2018 IPC Benchmarks"
_icaps-conference.org/ICAPS/2019/Workshop/WIPC_

### Official Review · AnonReviewer3 · 2019-04-18
**Interesting question but misses some deeper insights drawn from the result**

**Rating:** 5
**Confidence:** 4

**Review:**

The paper analyzes the benchmark set of the IPC 2018 and compares it to
the benchmarks of earlier IPCs. The analysis is done by computing
several features over all instances, normalizing the result and then
aggregating the values. The main conclusion seems to be that the
domains in IPC 2018 were more diverse than the ones in IPC 2014,
although this conclusion comes with a couple of caveats. There are not
much more conclusions drawn from the analysis and I'm not quite sure
what a reader should learn from the paper. For example, the domains
citycar and floortile are very similar to each other according to the
similarity measure defined in the paper but what does that tell us?
Will planners that work well on one domain always also work well on the
other? This question is not investigated, nor any other that goes
beyond defining a measure of similarity and reporting the values of
different domains according to it.

I'm not completely opposed to accepting the paper but I'm missing what
the title of the paper promises: some deeper insights that can be
drawn from the results.


Comments about the analysis:

Figures 1 and 2 are not clear. What is their main message? That in one
of the plot the different colors align more than in the other? This is
impossible to see because one color overlaps the other and the
normalized values vary so much anyway. Maybe sorting the features by
the normalized value in one of the problems would help a bit but even
then I don't see what the message of the Figure should be.

The normalized values in Figure 1 seem to be all very close to the
extremes which makes sense if the underlying distribution of values is
superlinear (like number of facts) but scaled linearly between the
minimum and maximum observed value. Normalizing linearly according to
the minimal/maximal observed value instead of according to the
underlying distribution can bias the results.

You mention that the difference measure can range from 0 to 107 (1 per
feature) but this is possible. Say you had a single feature and 3
problems that are as different to each other as possible. They would be
evenly spaced between 0 and 1 after normalization ([0, 0.5, 1]). Their
average pairwise distances would be (0 + 0.5 + 1 + 0.5 + 0 + 0.5 + 1 +
0.5 + 0) / 3^2 = 4/9 < 1. There is no way to get to 1 and the larger
the number of problems, the smaller the maximal value will be. I think
for 20 problems, the maximal value would be 0.35, so with 107 features,
the differences could only go as high as 37.5.



Minor comments:

I don't understand the sentence "deleted in the computation of the
relaxed plan". FF ignores delete effects, so its plans or their
computation doesn't delete facts.

There is no unique "landmark graph computed by Fast Downward". Fast
Downward has implementations for different ways of computing landmarks
that originate from different papers and authors. You should describe
the landmarks actually being used and cite the original paper for it.

ADL should be capitalized.

I don't see what "as we saw before" refers to.

The term "in a domain basis" is not clear.

"We sum all the rows (or columns) in the matrix" should probably be "we
sum all entries in the matrix".

$\sum f_i(p_k), \forall p_k \in D_k$ is not a correct way of writing a
sum. It should be $\sum_{p_k \in D_k} f_i(p_k)$.

---

### Official Review · AnonReviewer2 · 2019-04-25
**Insights from the 2018 IPC Benchmarks**

**Rating:** 7
**Confidence:** 4

**Review:**

Summary

This paper considers a number of features that the authors consider to determine the diversity of problem instances and analyses the problems used in the two most recent IPCs with respect to these features.  Overall, I think this paper will make a good contribution to the workshop and will lead to some interesting discussion.

Some points the authors might want to consider before the camera ready/discussion are:
1) It would be nice to see some more convincing discussion of why the selected features are considered to make the instances diverse (currently we have them, but there's no justification of why they might indeed define diversity of problems rather than anything else) and what other features might be informative but could not be considered (and why).  The authors do acknowledge this in future work, but some more insights into what they have learnt from this would be useful (and vital to anyone wanting to continue this line of work without repeating what has already been done) and some discussion up front about the reasoning for using these features and their potential limitations is needed.
2) It would be nice to see more analysis, particularly in the Intra and Inter Domain analyis sections: currently we have the results reported but not much insight:  little detail on what the features that make the difference are, and discussion of how this ties with competition results (I'm not convinced there's a reasonable argument for leaving an initial look at this to future work, given the results are available), and human intuition of whether these are indeed the more 'different' instances.  Even if this is speculation at this stage, it would be an interesting start for discussion.
3) The 'by domain' analysis is not so convincing, because it looks at features of the instances, rather than of the domains themselves, it would be interesting to have some discussion of this, and also of what features domains might have rather than instances (or indeed whether analysis without instances is even possible).  As well as some discussion of variation within domains versus variation between domains on instances.  Again the by-domain analysis would benefit from some similar insights to those suggested for the instances in point 2), do the differences align with the more intuitively 'different' domains, and the results of the performance of the planners in the competition.

The paper considers some interesting questions, and raises interesting issues.

Other Comments

It would be interesting to see some brief discussion on how/whether the issue of diversity is dealt with in the competitions of other communities (e.g. the SAT/ASP communities that are mentioned in the introduction).

In the future it might be intersting to analyse the domains over more years, to see whether diversity is increasing or decreasing.

Perhaps a controversial note but will a diverse range of domains not just necessarily favour portfolios even further?  Since necessarily a variety of techniques would then be even more likely to be the best solution.

I think a key debate in this paper, that isn't really touched upon until too late, is whether these features truly represent the difficulty of a given problem for a certain planner (although perhaps the success of IBACOP can go some way towards suggesting that they do, is there a citation, some insight to a paper on IBACOP where this was considered?).  There are some other features, that might be more difficult to detect, but might be more telling: for example, the presence of dead ends in the domain; ratios between heuristic values and actual plan lengths; particular types of domain that certain heuristics perform poorly on (e.g. is there a particular trap a delete relaxation heuristic might fall into), particular domains in which it's easy to take a wrong turn (perhaps those with dead ends again) that are suited to an LPG style local search (not really applicable to optimal planning here) versus a more conventional forward search etc.

As an aside, it's interesting to consider how this would translate to more expressive planning: numeric planning, temporal planning, hybrid planning; although actually the choice of planner here is often determined by which actually supports the necessary features (TILs, required concurrency, numeric effects); not by which will solve a problem with these features most quickly!

"We discarded organic-synthesis, since we only could extract the features of 5 problem instances within the time limit." I'm intrigued as to whether the features (or which features) are easily predictable from smaller instances: how does an analysis based on only smaller problems compare to one done on larger ones: do we need to be analysing the larger instances or not?

"Setting the feature value to the average of that feature values in the domain, if there exist at least one problem in the domain in which the feature has been successfully extracted." There are other ways you could continue doing this, for example, if looking for the number of objects (or any other feature), and this has increased on a per-instance basis then it might make sense to use the max rather than the average.

"For this purpose, we deleted the set of features which have the same value for all the problem instances of the competition." It would be interesting to know what these are, and indeed which other features had very similar values, and which were the most distinguishing.

It is interesting that the features are normalized, did you consider the relative importance of different features: I'm sure some are more valuable than others.

What would be helpful in the discussion of Table 3 is some sort of sense of scale, am I right in thinking that the maximum potential difference is 107 (or if it's difference from the mean approx half of this, if half of the instances scored 107 and half zero) but most domains differ by less than 5?  Some insight into how different 5 means would be helpful.  Currently we seem to be seeing a very small use of the range, is this because most features are very uniform?  Which features are particularly variable, and which are mostly uniform within each domain?  Are the top 10 most distinguishing similar across all domains, or very different?  The paper could actually add a whole new interesting dimension from the same results here: analysing the features, as well as analysing the problems themselves!

"Domains like parking and visitall in 2014, and spider or organic-synthesis in 2018 have very diverse problems, while all the instances in data-network or barman seem to be similar" How does this align with:
1) Your perception as knowledgeable people looking at the PDDL instances: would you say what you would consider more interestingly varied instance sets from your own perspective correspond to the results from the analysis?  Is there anything particularly interesting about these instances from a human perspective? And are there a few particular features that are setting the instances apart from each other, or is it a small gain across many features?
2) The results of the competition, are these domains where the instances are more different ones we see greater difference between the performance of planners?  Are there any domains where the instances are similar according to the metrics, but the same planner performs very differently across instances?

"The IPC 2018 has more diverse problems within the same domain, with an average difference of 5.6 against the average difference of 4.5 in the case of the IPC 2014." How siginficant is this given the range 0-107 you mentioned earlier, and a change from 4.5 to 5.6 (it seems like this could be almost entirely 1 (or 2) feature(s) that did this, given the order of magnitude of the difference is 1)?

Some of the features, e.g. number of objects, number of variables in causal graph etc. seem instance specific rather than domain specific: isn't it risky to compare diversity of domains using these metrics, given the results could change depending on what instances we see for those domains?

Typos
associated to -> associated with (multiple occurences)
SAS+ should be bold (top of page 2)
an special -> a special
To test how diverse are the problems of a domain -> To test how diverse the problems of a domain are